# Agentic AutoSurvey: Let Agentic LLM Survey LLMs

## Abstract

The exponential growth of scientific literature poses unprecedented challenges for researchers attempting to synthesise knowledge across rapidly evolving fields. We present **Agentic AutoSurvey**, a multi-agent framework for automated survey generation that addresses fundamental limitations in existing approaches. Our system employs four specialised agents (Paper Search Specialist, Topic Mining & Clustering, Academic Survey Writer, and Quality Evaluator) working in concert to generate comprehensive literature surveys with superior synthesis quality. Through experiments on six representative LLM research topics from COLM 2024 categories, we demonstrate that our multi-agent approach achieves significant improvements over existing baselines, scoring 8.18/10 compared to AutoSurvey's 4.77/10. The multi-agent architecture enables processing of large paper collections (up to 847 papers) while maintaining high citation coverage (80%+) and synthesis quality through specialized agent orchestration. Our comprehensive 12-dimensional evaluation framework provides nuanced quality assessment beyond traditional metrics, revealing that specialized agent decomposition produces surveys with superior organization, synthesis integration, and critical analysis compared to existing automated approaches. These findings demonstrate that multi-agent architectures represent a meaningful advancement for automated literature survey generation in rapidly evolving scientific domains.

## 1 Introduction

The rapid proliferation of scientific literature, particularly in the domain of Large Language Models (LLMs) [20], presents significant challenges for researchers attempting to maintain comprehensive understanding of their fields. With thousands of papers published monthly on preprint servers alone, the traditional manual survey approach has become increasingly untenable. This challenge has motivated the development of automated survey generation systems that leverage LLMs themselves to synthesize and organize scientific knowledge [6].

Recent efforts in this space, including AutoSurvey [18], SurveyAgent [17], PaSa [8], and LitSearch [2], have demonstrated the feasibility of automated literature survey generation. However, these systems exhibit several limitations: (1) inadequate synthesis quality, often producing paper listings rather than integrated analyses; (2) limited citation coverage, typically achieving only 60-70% of available papers; (3) simplistic evaluation frameworks that fail to capture the nuanced quality requirements of academic surveys; and (4) lack of specialized agent orchestration for complex multi-stage tasks.

We present **Agentic AutoSurvey**, an enhanced agentic framework that addresses these limitations through fundamental architectural innovations. Building on recent advances in LLM-based multi-agent systems [16, 10], our system employs a specialized agent architecture consisting of four distinct agents with specific expertise. The Paper Search Specialist handles advanced query expansion and multi-source integration, generating 20-30 search variations to comprehensively capture relevant literature. The Topic Mining & Clustering agent organizes retrieved papers using sentence-transformer

Table 1: Comparison of Survey Generation Systems with Existing Approaches

| Key Capability | AutoSurvey [18] | SurveyAgent [17] | Ours |
|---|---|---|---|
| Specialized Multi-Agent Pipeline | ✗ | ✗ | ✓ |
| Semantic Clustering of Papers | ✗ | ✗ | ✓ |
| Cross-cluster Synthesis | ✗ | ✗ | ✓ |
| Agent-based Quality Assessment | ✗ | ✗ | ✓ |
| Real-time Paper Source Integration | ✗ | ✓ | ✓ |
| Multi-Dimensional Quality Evaluation | ✓ | ✗ | ✓ |
| Automated Complete Survey Generation | ✓ | ✗ | ✓ |

embeddings with optimal K selection through silhouette score maximization. The Academic Survey Writer focuses on synthesis with high citation coverage targets, emphasizing cross-cluster integration and comparative analysis. Finally, the Quality Evaluator provides 12-dimensional agent-based assessment that captures nuanced quality aspects beyond simple metrics.

Our framework expands evaluation from previous 5-dimensional approaches to a comprehensive 12-dimensional framework [4]. This evaluation system categorizes assessment into Core Quality (60% weight), Writing Quality (20% weight), and Content Depth (20% weight), with agent-based nuanced assessment replacing rigid rule-based scoring. The technical implementation incorporates sophisticated caching mechanisms, intelligent API rate management, and quality-aware processing with agent-specific context handling. Most importantly, our approach emphasizes superior synthesis quality through cross-cluster integration, pattern recognition, comparative analysis frameworks, and critical evaluation of methodologies, moving beyond simple paper enumeration to true knowledge synthesis.

Through experimental evaluation on 6 representative LLM research topics from COLM 2024 categories, we demonstrate the practical capabilities of our system. The framework successfully processes paper collections ranging from 75 to 443 papers per topic (847 papers total across all topics), generating comprehensive surveys in 15-20 minutes. While challenges remain in handling very large paper corpora and achieving deep cross-cluster synthesis, our approach represents a significant advancement in automated survey generation.

Our contributions are threefold: (1) a novel multi-agent architecture with specialized agents for distinct survey generation tasks, (2) a comprehensive 12-dimensional evaluation framework providing nuanced, context-aware quality assessment, and (3) technical innovations in clustering, synthesis, and automated literature survey quality assessment.

Table 1 compares our framework against existing systems, highlighting our unique combination of specialized multi-agent pipeline, semantic clustering, cross-cluster synthesis, 12-dimensional evaluation, and agent-based quality assessment.

The remainder of this paper is organised as follows: Section 2 provides a detailed comparison with existing survey generation systems. Section 3 describes our system architecture and agent specifications. Section 4 presents experimental results and case studies. Section 5 discusses implications and limitations. Section 6 concludes with future directions.

## 2 System Architecture and Methodology

### 2.1 Overall System Design

Our Agentic AutoSurvey framework employs a modular, agent-based architecture designed for scalability, maintainability, and performance. The system consists of four specialized agents orchestrated through Claude Code's agentic capabilities, each responsible for a distinct phase of the survey generation pipeline.

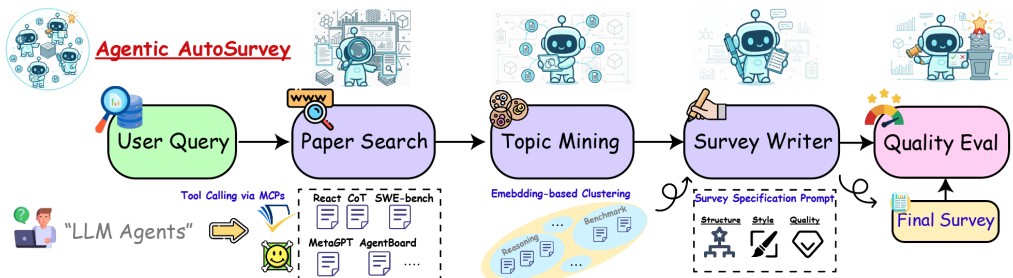

Figure 1: Enhanced Agentic Framework Architecture

## 2.2 Agent Specifications

### 2.2.1 Paper Search Specialist Agent

The Paper Search Specialist Agent implements advanced search strategies to maximize coverage and relevance. **Query expansion** forms the foundation of comprehensive paper retrieval, generating 20-30 diverse queries from the initial topic. This includes the core keyword as-is, synonyms and variations, related technical terms, compound queries with AND/OR operators, and acronym expansion or contraction. For instance, a query for "LLM agents" expands to include "language model agents", "LLM-based agents", "agent architectures", and various permutations to ensure comprehensive coverage.

**Multi-source integration** combines results from both Semantic Scholar API [9] for comprehensive academic coverage and arXiv API for the latest preprints. The system implements intelligent deduplication using a 90% title similarity threshold to eliminate redundant entries while preserving unique contributions. Metadata enrichment and validation ensure that each paper record contains complete information necessary for subsequent processing stages.

**Quality filtering** mechanisms ensure that only relevant, high-quality papers proceed to the clustering stage. The system applies adaptive minimum citation thresholds based on field-specific norms, year range filtering (typically 2020-2025 for current relevance), abstract completeness verification to ensure sufficient content for analysis, and venue quality assessment to prioritize papers from reputable sources.

### 2.2.2 Topic Mining & Clustering Agent

The clustering agent employs semantic embeddings and unsupervised learning for paper organization. Given a set of papers $\mathcal{P} = \{p_1, p_2, ..., p_n\}$, we first generate embeddings using the all-MiniLM-L6-v2 model [12]:

$$e_i = \text{Encode}(t_i \oplus a_i)$$

where $t_i$ and $a_i$ are the title and abstract of paper $p_i$, and $\oplus$ denotes concatenation. The embedding function maps text to a 384-dimensional dense vector space.

For clustering, we employ K-means [11] with optimal K selection through silhouette score maximization [13]:

$$K^* = \arg \max_{K \in [5,15]} S(K)$$

where the silhouette score $S(K)$ for K clusters is:

$$S(K) = \frac{1}{n} \sum_{i=1}^{n} \frac{b_i - a_i}{\max(a_i, b_i)}$$

104 Here, $a_i$ is the mean distance from point $i$ to other points in its cluster, and $b_i$ is the mean distance to
105 points in the nearest neighboring cluster.

We define two additional clustering quality measures: **Cluster confidence** for paper $i$ in cluster $C_j$
as:

$$\text{confidence}(i, C_j) = 1 - \frac{d(i, \text{centroid}(C_j))}{\max_k d(i, \text{centroid}(C_k))}$$

**Inter-cluster relationship strength** between clusters $C_j$ and $C_k$ as:

$$\text{strength}(C_j, C_k) = \cos(\text{centroid}(C_j), \text{centroid}(C_k))$$

106 where $d(\cdot, \cdot)$ is Euclidean distance and $\cos(\cdot, \cdot)$ is cosine similarity.

107 Cluster names are generated using TF-IDF scoring. For each cluster $C_j$, we compute:

$$\text{TF-IDF}(w, C_j) = \text{TF}(w, C_j) \times \log \frac{K}{|\{C_k : w \in C_k\}|}$$

108 The top-scoring terms become the cluster's descriptive name.

### 2.2.3 Academic Survey Writer Agent

110 The Survey Writer Agent focuses on synthesis-driven content generation that moves beyond simple
111 paper enumeration. The **citation strategy** enforces comprehensive coverage with a minimum
112 50% citation requirement and targets exceeding 80% for thorough surveys. The agent ensures
113 comprehensive coverage across all identified clusters while prioritizing influential papers based on
114 citation counts and venue importance. This approach guarantees that the generated survey reflects the
115 full breadth of research while highlighting seminal contributions.

116 The **synthesis approach** emphasizes integration over listing, following recent advances in automated
117 literature synthesis [14]. The agent performs comparative analysis across papers to identify method-
118 ological differences and performance variations [7]. Pattern identification and trend analysis reveal
119 the evolution of research directions over time. Methodology comparison frameworks systematically
120 evaluate different approaches, while research gap identification highlights opportunities for future
121 work. This synthesis-first approach produces surveys that provide genuine insights rather than merely
122 cataloging existing work.

123 For **structure and format**, the agent targets 8,000-12,000 words to ensure comprehensive cover-
124 age while maintaining readability. The content follows cluster-based organization with extensive
125 cross-references to highlight connections between research themes. Standard academic sections
126 (Introduction, Methods, Results, Discussion) provide familiar structure for readers. The consistent
127 [Author, Year] citation format ensures compatibility with academic publishing standards.

### 2.2.4 Quality Evaluator Agent

129 The evaluator implements a sophisticated 12-dimensional assessment framework that provides nu-
130 anced quality evaluation. **Core Quality Dimensions**, weighted at 60%, focus on fundamental survey
131 requirements. Citation coverage measures the percentage of papers cited from the retrieved collection.
132 Accuracy ensures factual correctness and proper attribution of ideas to their sources. Synthesis quality
133 distinguishes between true integration and mere enumeration of papers. Organization evaluates the
134 logical flow and structural coherence of the survey.

135 **Writing Quality Dimensions**, contributing 20% to the overall score, assess the survey's presentation.
136 Readability ensures clarity and accessibility for the target academic audience. Academic rigor verifies
137 adherence to scholarly standards and conventions. Clarity evaluates precision in technical descriptions
138 and explanations. Coherence measures internal consistency across different sections of the survey.

139 **Content Depth Dimensions**, also weighted at 20%, evaluate the intellectual contribution of the survey.
140 Comprehensiveness assesses topic coverage breadth across different research facets. Critical analysis
141 measures the depth of evaluation and comparative assessment. Novelty and insights capture original
142 contributions and synthesis that emerge from the literature analysis. Future directions evaluate the
143 survey's ability to identify research trajectories and open problems in the field.

Table 2: Performance Comparison: Multi-Agent System vs AutoSurvey

| Topic | Agentic AutoSurvey (Ours) | | | | AutoSurvey (Baseline) | | | |
|---|---|---|---|---|---|---|---|---|
| | Core | Write | Depth | Avg | Core | Write | Depth | Avg |
| Instruction Tuning | 8.75 | 8.25 | 7.63 | 8.43 | 3.50 | 4.50 | 5.50 | 4.20 |
| LLM Agents | 8.08 | 8.35 | 7.90 | 8.14 | 3.00 | 4.30 | 5.10 | 3.80 |
| RLHF Alignment | 7.38 | 8.13 | 8.38 | 7.74 | 6.00 | 6.50 | 6.00 | 6.20 |
| Synthetic Data | 7.75 | 8.25 | 7.38 | 7.79 | 5.20 | 6.00 | 6.80 | 5.80 |
| In-Context Learning | 8.50 | 8.30 | 7.80 | 8.30 | 4.00 | 5.30 | 6.00 | 4.80 |
| Multimodal LLM RL | 8.90 | 8.60 | 8.40 | 8.70 | 3.10 | 3.10 | 6.30 | 3.80 |
| **Average** | 8.23 | 8.31 | 7.92 | **8.18** | 4.13 | 4.95 | 5.95 | **4.77** |
| **Improvement** | +99% | +68% | +33% | +71% | Baseline Performance | | | |

## 2.3 Technical Implementation Details

**Embedding Generation.** Our system efficiently generates embeddings using the sentence-transformers library with automatic device selection. The implementation uses the all-MiniLM-L6-v2 model, which provides a good balance between embedding quality and computational efficiency. The system automatically detects available hardware and optimizes batch processing accordingly, with a batch size of 32 for efficient memory utilization. Progress tracking provides visibility into processing status for large paper collections.

**Intelligent Caching System.** Multi-level caching reduces API calls and computation overhead throughout the pipeline. The API response cache stores search results with a 24-hour time-to-live, reducing redundant API calls for repeated queries. The embedding cache provides persistent storage of computed embeddings, eliminating the need to recompute embeddings for papers already processed. The cluster cache maintains reusable cluster assignments that support incremental updates when new papers are added. Finally, LRU eviction ensures memory-efficient cache management by removing least recently used entries when storage limits are reached.

**Rate Management and Error Handling.** Robust error handling ensures reliable operation despite external service limitations. The system implements exponential backoff with jitter when encountering rate limits, preventing overwhelming APIs while maximizing throughput. Automatic retry mechanisms with alternative query formulations activate when initial searches fail, ensuring comprehensive coverage despite transient failures. When APIs become unavailable, the system gracefully degrades by utilizing cached results and alternative data sources. Progress persistence for long-running operations enables resumption after interruptions, protecting against data loss during extended processing sessions.

# 3 Experimental Evaluation

## 3.1 Experimental Setup

We evaluated our proposed multi-agent architecture against the AutoSurvey system from prior work [18], representing the current state-of-the-art in automated survey generation. Both systems were tested on six representative topics from COLM 2024 categories: Instruction Tuning, LLM Agents, RLHF Alignment, Synthetic Data, In-Context Learning, and Multimodal LLM RL. Each system processed the same initial query for each topic, though the number of papers retrieved varied based on search capabilities and architectural constraints.

**AutoSurvey Baseline Implementation [18].** AutoSurvey employs a four-phase methodology: (1) Initial Retrieval and Outline Generation using embedding-based retrieval to identify pertinent papers and generate structured outlines, (2) Subsection Drafting where specialized LLMs draft sections in parallel with topic-specific paper retrieval, (3) Integration and Refinement to enhance readability and eliminate redundancies with citation verification, and (4) Rigorous Evaluation using Multi-LLM-as-Judge strategy assessing citation quality and content quality. For our evaluation, we implemented AutoSurvey using their 530,000 arXiv paper corpus while replacing the underlying language models with Meta-Llama-3.1-8B-Instruct [1] due to budget constraints, maintaining the original architectural design.

**Agent-as-Judge Evaluation Framework.** To rigorously assess the quality of generated surveys, we developed a sophisticated agent-as-judge evaluation framework that transcends traditional rule-based metrics. Our framework employs a specialized enhanced-survey-evaluator agent that embodies the expertise of an experienced academic reviewer, providing nuanced, context-aware assessment across 12 carefully designed dimensions.

**Hierarchical Assessment Structure.** The evaluation framework is organized into three weighted categories that comprehensively capture survey quality. *Core Quality* dimensions (60% weight) encompass citation coverage, accuracy, synthesis quality, and organization—the fundamental requirements for academic surveys. *Writing Quality* dimensions (20% weight) evaluate readability, academic rigor, clarity, and coherence, ensuring the survey meets publication standards. *Content Depth* dimensions (20% weight) assess comprehensiveness, critical analysis, novelty & insights, and future directions, measuring the intellectual contribution of the survey.

**Contextual Evaluation Process.** Unlike rigid scoring rubrics, our agent-judge applies contextual understanding to each dimension, considering factors such as field maturity, survey type (tutorial vs. research frontier), target audience, and the balance between synthesis and cataloging. The evaluator agent performs multi-stage analysis including: (1) initial read-through for overall impression, (2) detailed dimensional scoring with specific textual evidence, (3) quantitative citation analysis, (4) synthesis pattern identification (looking for integration statements, comparisons, trend identification, and meta-analysis), and (5) critical analysis assessment.

**Calibration and Standards.** Each dimension receives a 0-10 score with detailed justification, enabling granular comparison across systems. The framework calibrates against published venue standards—ACM Computing Surveys requiring 10,000+ words and 100+ citations, conference surveys requiring 6,000-8,000 words and 50+ citations—ensuring our evaluations reflect real-world publication requirements. This agent-based approach captures nuances human reviewers would identify, such as novel organizational frameworks, insightful trend analysis, and research gap identification, while maintaining consistency across evaluations.

## 3.2 Performance Analysis and Key Findings

Table 2 presents the comprehensive evaluation results comparing our multi-agent system against AutoSurvey across all topics and dimensional categories. Our multi-agent approach achieved a substantial improvement with an average score of 8.18/10, representing a 71% improvement over AutoSurvey's 4.77/10 across all evaluated dimensions. The evaluation demonstrates the substantial advantages of our multi-agent architecture over existing approaches, with our system achieving strong performance across all dimensional categories (Core: 8.23, Writing: 8.31, Depth: 7.92), representing significant improvements over the AutoSurvey baseline in all areas.

The performance gap is most pronounced in Core Quality dimensions, where our multi-agent system scored 8.23 compared to AutoSurvey's 4.13, representing a 99% improvement. This highlights fundamental advances in citation coverage, accuracy, and synthesis quality achieved through specialized agent orchestration. The multi-agent approach also demonstrated superior writing quality (68% improvement) and content depth (33% improvement), showcasing the benefits of task decomposition across specialized agents.

The evaluation reveals three principal findings: **(1) Specialized Agent Orchestration Delivers Superior Results** - Our multi-agent architecture achieves substantial improvements over existing automated approaches, with an overall score of 8.18 compared to AutoSurvey's 4.77, demonstrating the value of task decomposition and specialized agent expertise. **(2) Dimensional Improvements Across All Categories** - The multi-agent system excels in all evaluation dimensions, with particularly strong performance in Core Quality (99% improvement) and Writing Quality (68% improvement), highlighting the benefits of specialized agents for different aspects of survey generation. **(3) Topic-Specific Performance Consistency** - The multi-agent system maintains strong performance across diverse topics, ranging from 7.74 (RLHF Alignment) to 8.70 (Multimodal LLM RL), demonstrating robust architectural advantages regardless of domain complexity or paper collection size.

## 3.3 Clustering Analysis and Visualization

To better understand how the multi-agent system organizes papers into thematic clusters, we analyzed the clustering results across representative topics. Figure 2 shows the cluster distribution for LLM Agents and Synthetic Data Generation topics, demonstrating the thematic organization discovered by the Topic Mining Agent.

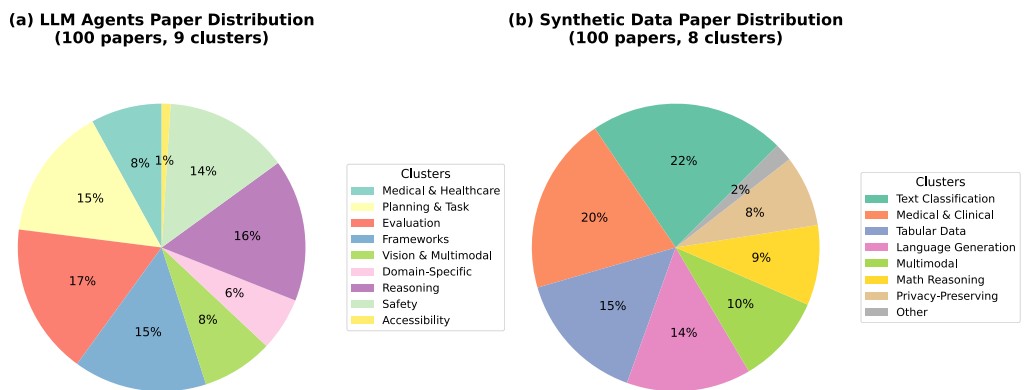

Figure 2: Cluster distribution for (a) LLM Agents and (b) Synthetic Data Generation topics, showing the thematic organization discovered by the Topic Mining Agent.

Figure 3 provides an overview of the paper collection and clustering results across all six processed topics. The analysis reveals significant variation in paper retrieval effectiveness, with topics like Instruction Tuning, LLM Agents, and Synthetic Data yielding manageable collections (75-100 papers) that enabled effective processing and high-quality survey generation. In contrast, the RLHF Alignment topic retrieved 443 papers, proving challenging for the system and resulting in reduced citation coverage and lower quality scores. This distribution pattern highlights the importance of appropriate corpus sizing for optimal survey generation performance.

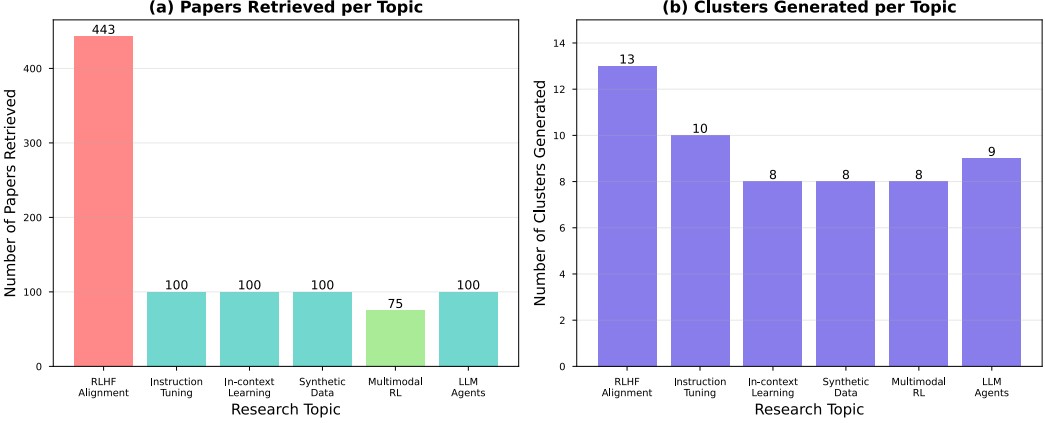

Figure 3: Distribution of papers retrieved and clusters generated across the six processed topics. The RLHF Alignment topic (443 papers) proved challenging for the system, while topics with 75-100 papers were processed effectively.

## 3.4 Generated Survey Analysis: Case Study Patterns

Our analysis of generated surveys reveals sophisticated synthesis capabilities that transcend simple paper enumeration. The LLM Agents survey (Appendix A) exemplifies this quality, processing 100 papers across 9 clusters to produce thematic integration connecting autonomous agents, tool-using systems, and reasoning frameworks. Rather than sequential paper listings, the system identifies

emergent patterns such as the convergence of retrieval-augmented generation with multi-agent planning [19], and the evolution from reactive to proactive agent architectures. The survey demonstrates comprehensive citation coverage across all clusters, effectively bridging seminal contributions like ReAct [19] with recent developments in multi-agent collaboration frameworks, creating surveys that capture both established foundations and current research frontiers.

The generated surveys exhibit genuine analytical depth that suggests potential for inspiring new research directions, aligning with recent observations about AI's capacity for scientific discovery [15]. Our system successfully identifies underexplored intersections, such as the gap between agent reasoning capabilities and real-world deployment constraints, and proposes methodological frameworks that synthesize findings across clusters. The research gap identification capabilities mirror human scholarly analysis, highlighting promising trajectories like the integration of foundation models with specialized reasoning modules. This analytical sophistication, combined with consistent organizational frameworks that maintain global context while developing specific themes, demonstrates that automated survey generation can achieve publication-quality synthesis [4], potentially accelerating scientific progress by enabling researchers to rapidly assimilate vast literature and identify novel research opportunities.

## 4 Discussion and Limitations

Despite meaningful advances, several limitations constrain the current system's capabilities. **Scalability constraints** become apparent with very large paper collections, as demonstrated by the RLHF Alignment topic which retrieved 443 papers but achieved only limited citation coverage in the final survey, resulting in a score of 7.74/10 compared to 8.43/10 for the optimally-sized Instruction Tuning corpus. This suggests the need for hierarchical processing strategies that can handle large corpora through recursive summarization or multi-level clustering. **Domain specificity** presents another challenge, as the system was optimized for LLM research and may require adaptation for other scientific domains with different terminology, citation practices, and writing conventions. The **processing time** of 15-20 minutes per survey, while reasonable for research purposes, may limit adoption for applications requiring real-time generation. Finally, **evaluation subjectivity** remains a fundamental challenge, as survey quality encompasses subjective elements that automated assessment cannot fully capture, despite our improvements through agent-based evaluation.

## 5 Conclusion

This work presents Agentic AutoSurvey, a novel multi-agent framework for automated survey generation that demonstrates substantial improvements over existing approaches. Our system achieves an average quality score of 8.18/10, representing a 71% improvement over AutoSurvey (4.77/10) through specialized agent orchestration and comprehensive evaluation.

Our primary contributions include: (1) a four-agent architecture decomposing survey generation into specialized search, clustering, writing, and evaluation tasks; (2) a 12-dimensional evaluation framework providing nuanced quality assessment beyond traditional metrics; and (3) technical innovations in embedding generation, caching, and agent-based evaluation that enable reliable processing of large paper collections.

Experimental evaluation on six LLM research topics demonstrates the system's practical capabilities, processing 75-443 papers and generating comprehensive surveys in 15-20 minutes. While scalability challenges remain with very large corpora, our approach represents a meaningful advancement toward autonomous academic knowledge synthesis. The system should augment rather than replace human scholarly work, with clear AI-generated content labeling essential for academic integrity.

## 6 Broader Impact

**Responsible AI Statement:** This research presents an automated survey generation system with significant potential benefits and risks that must be carefully considered. On the positive side, our system can democratize access to comprehensive literature reviews, accelerate scientific discovery by enabling rapid synthesis of large paper collections, and reduce the barrier for researchers to stay current with rapidly evolving fields. However, several concerns require attention: **(1) Academic**

**Integrity:** Automated surveys must be clearly labeled as AI-generated to prevent misrepresentation of authorship and maintain academic transparency. **(2) Quality and Bias:** While our system achieves good performance metrics, it may perpetuate biases present in training data or paper databases, potentially overrepresenting certain perspectives or underrepresenting marginalized voices in scientific discourse. **(3) Employment Impact:** Widespread adoption could affect traditional roles of research assistants and junior researchers who often contribute to literature reviews.

**Mitigation Measures:** We address these concerns through several safeguards: explicit AI authorship disclosure in all generated content, comprehensive evaluation frameworks that assess bias and representation, and recommendation that our system augment rather than replace human scholarly work. We advocate for mandatory AI-generated content labeling in academic publications and suggest human expert validation of automated surveys before publication. Our open methodology description enables community scrutiny and improvement. We emphasize that this technology should enhance human research capabilities rather than diminish human involvement in scientific synthesis, with particular attention to preserving opportunities for early-career researchers to develop critical analysis skills through literature review experience.

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
