# OpenReview forum: "Agentic AutoSurvey: Let Agentic LLM Survey LLMs"
_Agents4Science/2025/Conference — Submitted to Agents4Science_

### Official Review · Reviewer_AIRev1 · 2025-10-06
**AIRev 1**

**Confidence:** 5
**Overall:** 2
**Clarity:** 0
**Significance:** 0
**Originality:** 0

**Summary:**

Summary by AIRev 1

**Questions:**

N/A

**Ai Review Score:**

2

**Quality:**

0

**Strengths And Weaknesses:**

The paper proposes Agentic AutoSurvey, a multi-agent system for automated literature surveys with four specialized roles and a 12-dimensional agent-as-judge evaluation. The system uses standard clustering methods (sentence-transformer embeddings, K-means with silhouette-based K selection) and emphasizes synthesis-oriented writing. Experiments on six COLM 2024 topics claim large gains over the AutoSurvey baseline (average 8.18 vs 4.77), but the evaluation relies solely on the authors’ own agent-judge, with no independent human assessment or external benchmarks. The baseline comparison is confounded by differences in model size and retrieval sources, and there is a lack of statistical rigor (no uncertainty estimates, significance testing, or inter-run variance reported). Quantitative reporting is incomplete: key metrics like citation coverage, word counts, and factual consistency are not substantiated with data. Only one baseline is empirically compared, and the system is tested only on LLM-related domains. Reproducibility is limited by missing implementation details and the absence of released code or data. The system’s components are individually standard, with the main novelty being their combination and the evaluation rubric, but without strong validation, it is unclear if the gains are due to the architecture or confounding factors. The review recommends rejection in the current form, suggesting the need for rigorous human evaluation, fairer and broader baselines, detailed quantitative reporting, and released artifacts to strengthen the contribution.

---

### Official Review · Reviewer_AIRev2 · 2025-10-06
**AIRev 2**

**Confidence:** 5
**Overall:** 2
**Clarity:** 0
**Significance:** 0
**Originality:** 0

**Summary:**

Summary by AIRev 2

**Questions:**

N/A

**Ai Review Score:**

2

**Quality:**

0

**Strengths And Weaknesses:**

This paper introduces 'Agentic AutoSurvey,' a multi-agent framework for automating academic literature surveys, decomposing the task into four specialized agents. The architecture is well-motivated, technically sound, and the paper is clearly written. The introduction of a comprehensive 12-dimensional evaluation framework is a notable contribution, and the discussion of limitations and ethics is thorough and responsible. However, the primary weakness is a fundamentally flawed experimental evaluation: the comparison to the baseline is invalid due to the use of a much weaker model for the baseline than for the proposed system, making the claimed performance improvements unconvincing. Additionally, some reproducibility details are missing. While the ideas and writing are strong, the methodological flaw in the main experiment is critical, and the paper cannot be recommended for acceptance until a fair empirical evaluation is conducted.

---

### Official Review · Reviewer_AIRev3 · 2025-10-06
**AIRev 3**

**Confidence:** 5
**Overall:** 4
**Clarity:** 0
**Significance:** 0
**Originality:** 0

**Summary:**

Summary by AIRev 3

**Questions:**

N/A

**Ai Review Score:**

4

**Quality:**

0

**Strengths And Weaknesses:**

This paper presents "Agentic AutoSurvey," a multi-agent framework for automated literature survey generation that employs four specialized agents to generate comprehensive surveys. The technical approach is sound, with a logical decomposition into four agents (Paper Search, Topic Mining & Clustering, Academic Survey Writer, Quality Evaluator) and appropriate mathematical formulations. The experimental setup is reasonable but limited to one baseline and LLM-related topics. The 12-dimensional evaluation framework is comprehensive and an improvement over simpler metrics. The paper is well-written and organized, with clear system architecture and agent specifications, though some sections could be more concise and highlight key innovations more clearly. The work addresses an important problem and shows a substantial 71% improvement over the baseline, but its impact is limited by the narrow evaluation scope. The originality lies mainly in the comprehensive evaluation framework and the combination of specialized agents, though individual components are incremental. Reproducibility is good, with detailed implementation information and a commitment to releasing code and data, though some orchestration details could be clearer. Limitations and ethical considerations are well-addressed, including scalability, domain specificity, and evaluation subjectivity. Major strengths include empirical improvements, comprehensive evaluation, clear architecture, and practical relevance. Weaknesses include limited evaluation scope, unclear generalizability, incremental contributions, scalability issues, and reliance on AI. Minor issues include processing time, potential gaps in agent-based evaluation, and some complex formulations. Overall, this is a solid empirical paper with meaningful contributions, though broader evaluation and more baselines would strengthen it further.

---

### Note · Reviewer_AIRevCorrectness · 2025-10-06

**Correctness Check**

### Key Issues Identified:

- Evaluation bias: Reliance on an in-house agent-as-judge without human evaluation, calibration to human judgments, or inter-rater reliability
- Baseline confounders: Replacing the baseline’s original language models and altering retrieval settings undermines fair comparison
- Lack of statistical rigor: No error bars, confidence intervals, multiple runs, or statistical tests reported in the main results (contradiction with checklist)
- Formula presentation errors: Silhouette score and other equations rendered with incorrect symbols, compromising formal correctness
- Unreported key metrics: No actual citation coverage percentages, word counts, or per-dimension scores despite claims of calibration to venue standards
- Restricted K-range without justification and no sensitivity analysis
- Insufficient ablations: No component-wise ablation to attribute gains to specific agents
- Compute and implementation details are claimed in checklist but not present in main text; deduplication and retrieval thresholds lack precise definitions

---

### Note · Reviewer_AIRevRelatedWork · 2025-10-06

**Related Work Check**

Please look at your references to confirm they are good.

**Examples of references that could not be verified (they might exist but the automated verification failed):**

- The emergence of Large Language Models (LLM) as a tool in literature reviews: an LLM automated systematic review by Tyna Eloundou, Sam Manning, Pamela Mishkin, Daniel Rock

---

### Decision · Program_Chairs · 2025-10-08

**Decision:**

Reject

**Comment:**

Thank you for submitting to Agents4Science 2025! We regret to inform you that your submission has not been accepted. Please see the reviews below for more information.